Effects of phosphorus-solubilizing fungi on bulb quality and the Fritillaria taipaiensis rhizosphere soil environment

Ren Xueyang 1 2
Yuan Lin 2
Yao Huan 1 2
Wang Yuhan 1 2
Wang Huan 2
Guo Dongqin 3 guodongqin1997@163.com
Zhou Nong 1 2 erhaizn@126.com
1 Chongqing Engineering Research Centre for Green Cultivation and Deep Processing of Taoist Medicinal Herbs in the Three Gorges Reservoir Area , Chongqing , China
2 School of Biology and Food Engineering, Chongqing Three Gorges College , Chongqing, Wanzhou , China
3 School of Pharmacy, Chongqing Three Gorges Medical College , Chongqing, WanZhou , China
Nikalje Ganesh
Electronic publication date: 2025 Apr 21
Publication date: 2025
Volume: 13
Electronic Location ID: e19283
Received 2024 Oct 21; Accepted 2025 Mar 17
Copyright: © 2025 Ren et al.
Copyright year: 2025
Copyright holder: Ren et al.
License: This is an open access article distributed under the terms of the Creative Commons Attribution License, which permits unrestricted use, distribution, reproduction and adaptation in any medium and for any purpose provided that it is properly attributed. For attribution, the original author(s), title, publication source (PeerJ) and either DOI or URL of the article must be cited.
License URL: https://creativecommons.org/licenses/by/4.0/

Keywords: Phosphorus-soluble fungi, Biomass, Drug quality, Rhizosphere soil, Fritillaria taipaiensis P.Y.Li

Funding: Chongqing Natural Science Foundation project CSTB2023NSCQ-LMX0010 Chongqing Key Laboratory of Development and Utilization of Genuine Medicinal Materials in Three Gorges Reservoir Area KFKT2022014 This work was supported by the Chongqing Natural Science Foundation project (CSTB2023NSCQ-LMX0010); Open Project Program of Chongqing Key Laboratory of Development and Utilization of Genuine Medicinal Materials in Three Gorges Reservoir Area (Grant No. KFKT2022014). The funders had no role in study design, data collection and analysis, decision to publish, or preparation of the manuscript.

==============================
Background

Fritillaria taipaiensis P.Y. Li is commonly used in Chinese medicine for its cough-suppressing and expectorant properties. Due to over-excavation and ecological damage, the wild resources of F. taipaiensis have suffered serious damage. Understanding and improving the inter-root soil environment plays an important role in improving the success rate of artificial cultivation of F. taipaiensis and the quality of medicinal herbs.

Methods

This study employed a pot experiment to inoculate three strains of phosphorus-solubilizing fungi from the Aspergillus genus for a total of seven treatment groups, with sterile physiological saline serving as the control group (CK). The research aims to examine the impact of inoculating phosphorus-solubilizing fungi on the biomass of F. taipaiensis, alkaloid concentration in its bulbs, and characteristics of the rhizosphere soil environment. The specific inoculation treatments included: Aspergillus tubingensis (Z1); Aspergillus niger (Z2); Aspergillus fumigatus (Z3); a combination of A. tubingensis and A. niger (Z12); a combination of A. niger and A. fumigatus (Z13); a combination of A. tubingensis and A. fumigatus (Z23); and a combination of all three fungi, A. tubingensis, A. niger, and A. fumigatus (Z123).

Results

Inoculation with phosphorus-soluble fungi significantly increased the biomass of F. taipaiensis, and the largest increase was in the Z123 group, which was 62.85% higher than that of the CK group. Total alkaloid content increased the most (0.11%) in the Z3 group, which was an 83.87% increase compared with the CK group. The total content of monomer alkaloids in the Z3, Z13, and Z123 groups increased by 10.53%, 12.48%, and 9.61%, respectively, compared with those in the CK group, indicating that the quality of F. taipaiensis could be significantly improved after applying phosphorus-solubilizing fungi. The soil environment improved after inoculation with different phosphorus-solubilizing fungi. The Z23 and Z123 groups had the greatest effect on the rhizosphere soil bacteria and Actinomyces. Overall, the soil nutrient content of the Z13 group increased the most, and the contents of available phosphorus, available potassium, available nitrogen, total phosphorus, and organic matter increased by 47.71%, 27.36%, 26.78%, 25.13%, and 31.72%, respectively, compared with those in the CK group.

Conclusion

These results show that the treatment groups that included different combinations of strains were superior to the single-strain treatment groups, and the Z123 group was the best treatment group when considering bulb biomass and alkaloid and soil nutrient contents. Applying phosphorus-solubilizing fungus fertilizer is highly feasible during F. taipaiensis production in the field.

Introduction

Fritillaria taipaiensis (Family Liliaceae) is a perennial herb. Their dried bulbs are used as medicine (Li et al., 2021b) and have a bitter, sweet, and slightly cold taste. This herb is used to resolve phlegm and relieve cough and asthma (Luo et al., 2021b) and has high medicinal value. It is a rare and authentic medicinal material in the Qinba Mountains. It was first recorded in the Chinese Pharmacopoeia in 2010 and is one of the fundamental species of Fritillaria. As one of the most successful F. fritillaria artificial breeding products, F. taipaiensis has natural advantages, such as strong adaptability, mature production technology, high yield, good quality, and low altitude cultivation. Small-scale cultivation areas have formed in the Sichuan, Shaanxi, and Chongqing areas (Fu, 2012). The heavy use of chemical fertilizers during the cultivation of medicinal materials causes a serious imbalance of nutrients in the soil, resulting in a lack of trace elements, excessive heavy metals, and a decrease in the quality of medicinal materials, which seriously affects the harvest of high-quality F. taipaiensis (Wei et al., 2021). Microbial fertilizer is an environmentally friendly product that has been widely used in the production of Chinese medicinal materials. It regulates the growth of crops and improves the soil environment and quality of medicinal materials (Dong, 2018).

Phosphorus is an indispensable nutrient element for plant growth and development (Liu et al., 2023b). In soil, phosphorus mainly exists in insoluble organic or inorganic forms, which are difficult for plants to directly absorb and utilize (Meng, 2022). The production of plant phosphatases occurs through various pathways, and phosphate-solubilizing fungi, as fundamental functional microorganisms in the phosphorus cycle, can produce phosphatases that convert insoluble phosphorus into forms that are utilizable by plants. This process allows plants to utilize large amounts of insoluble phosphorus fixed in soil minerals (Arias et al., 2023). Some scholars have applied phosphorus-solubilizing fungi to promote plant growth and improve crop yield and quality after inoculation in the production of Masson pine (Lü, 2021), peony (Xue et al., 2018), and Angelica (Jiang et al., 2023). The Aspergillus genus, a prevalent group of fungi, holds substantial importance across various sectors including the food industry, plant-soil ecology, and medicine. Studies have investigated the impact of interspecies chemical signaling from the phyllosphere symbiont Aspergillus cvjetkovicii on rice disease resistance and uncovered a defensive strategy against phyllosphere plant pathogens (Fan et al., 2024). Additionally, Mattos et al. (2021) identified the endophytic strain Aspergillus niger 9-P, isolated from indigenous forage grass, as a promising candidate for biofertilizers with plant growth promotion capabilities. This indicates that phosphorus-solubilizing fungi can alter soil microbial communities, improve soil fertility, enhance plant phosphorus absorption and utilization rates, and increase crop yield and quality. Therefore, we hypothesize that phosphate-solubilizing fungi may exert similar growth-promoting effects on F. taipaiensis and enhance the content of its medicinal components.

F. taipaiensis serves as a significant source of various medicinal active compounds (Yan et al., 2022), particularly steroidal alkaloids, which exhibit beneficial and therapeutic effects in phlegm resolution, cough alleviation, and asthma management (Li et al., 2024). Alkaloids represent the primary active constituents of F. taipaiensis and are critical indicators for assessing its quality. Microorganisms, which are integral components of the soil microenvironment, exhibit functional diversity that is intricately linked to soil quality and crop development (Duchene, Vian & Celette, 2017). Notably, soil fungi and bacteria contribute to the synthesis of soil humus, decomposition of organic matter, and regulation of soil carbon and nutrient cycles (Razgulin & Voronin, 2014; Liu et al., 2023a). Actinomycetes are capable of producing various enzymes that facilitate the breakdown of macromolecular substances, including cellulose, lignin, pectin, and proteins within the soil. Additionally, certain actinomycetes can synthesize antibiotics that inhibit plant pathogens, thereby promoting plant health (Javed et al., 2021). Phosphate-solubilizing fungi enhance phosphorus availability in the soil through the process of phosphate solubilization, which in turn supports plant growth (Fitriatin et al., 2022). Furthermore, soil pH influences the decomposition rate of organic matter, thereby affecting soil fertility and the growth and development of F. taipaiensis. The nutrient content of the soil also impacts the population of soil microorganisms and quality of F. taipaiensis.

Previous research on the beneficial effects of phosphate-solubilizing fungi on plant growth predominantly emphasized food crops, such as soybean (Ren, 2021) and wheat (Li et al., 2021a). In contrast, investigations into the influence of these fungi on the growth and development of medicinal plants have remained limited. Specifically, there is a paucity of studies addressing the effects of phosphate-solubilizing fungi on the growth and development of F. taipaiensis, a medicinal species. This study aims to utilize F. taipaiensis as the model organism and various phosphate-solubilizing fungi as experimental subjects to examine their potential impacts on the biomass, medicinal quality, rhizosphere soil microbial populations, and soil nutrient content of F. taipaiensis. Furthermore, the research will evaluate the differences across the various fungi. A comprehensive assessment of multiple indicators was conducted to identify the most effective phosphate-solubilizing fungi for enhancing the growth of F. taipaiensis, thereby providing a theoretical foundation for the application of these fungi in the cultivation of this medicinal plant.

Materials and Methods

Experimental material

Trial material

The F. taipaiensis seed source consisted of 3-year bulbs from Hongchiba (N 31°32′32.09″, E 109°4′ 57.76″) planting base in Wuxi County, Chongqing. The soil for the pot test was obtained from Chongqing Three Gorges University (N 108°45 ′36.87″ E 30°75′ 57.69″). The base soil, river sand, and organic fertilizer (2:1:1) were mixed on campus, passed through an 8-mm soil screen, intermittently sterilized for 2 h at 121 °C, sealed after natural cooling, and stored. The strains utilized in this study were obtained from soil samples collected from the roots of F. taipaiensis across various geographical locations. Initial screening for phosphorus solubilization potential was performed using the phosphorus solubilization circle method. This was followed by a more detailed assessment employing the molybdenum-antimony colorimetric technique to ascertain strains exhibiting superior phosphorus solubilization efficiency. The identification of these strains integrated morphological characteristics of the colonies with DNA sequence amplification of the internal transcribed spacer (ITS) region. Furthermore, the phosphorus solubilization capabilities of the selected strains were evaluated in liquid media containing diverse phosphorus sources. Ultimately, following this comprehensive screening and identification process, three exemplary strains were identified: A. tubingensis, A. niger, and A. fumigatus.

Experimental apparatus

The ME204E 1:10,000 analytical balance (METTLER Toledo Instrument Co., Ltd., Greifensee, Switzerland), ZXGP-A2270 ten-stage programming water-proof incubator (Shanghai Zhicheng Analytical Instrument Manufacturing Co., Ltd., Shanghai, China), BIOBASE type high-pressure steam sterilization pot (Shandong Boke Biological Industry Co., Ltd., Shandong, China), ZD-85 type gas bath constant temperature oscillator (Shanghai Xinuo Instrument Group Co., Ltd., Shanghai, China), SW-CJ-2F type ultra-clean workbench (Suzhou Antai Air Technology Co., Ltd., Jiangsu, China), and a UV-1200 type UV-visible spectrophotometer (Shanghai Medical Instrument Co., Ltd., Shanghai, China) were used in this study.

Chemicals and ingredients

The experimental reagents utilized in this study are presented in Table 1. These included chloroform, methanol, ammonium acetate, potassium dichromate, concentrated sulfuric acid, ammonium molybdate, o-phenanthroline, sodium bicarbonate, sodium hydroxide, and boric acid, all of which are of analytical grade. Additionally, methanol, acetonitrile, and diethylamine were classified as chromatographic grade.

Table 1 Directory of experimental chemicals and their suppliers.

Identification and references of the experimental pharmacological agents utilized.

Name of culture medium/reagent	Supplier name	Location of supplier	
Tiger Red Medium, LB Medium, Modified Gause No. 1 Medium, Inorganic Phosphate Medium	Qingdao High-Tech Industrial Park Haibo Biotechnology Co., Ltd.	Qingdao, Shandong, China	
Peimine, Peiminine, Peimisine, Sipeimine, Sipeimine-3-β-D-glucoside	Chengdu Zhibiao Huachun Biotechnology Co., Ltd.	Chengdu, Sichuan, China	
Potassium dichromate, Ammonium molybdate, Ammonium acetate	Chengdu Jinshan Chemical Reagent Co., Ltd.	
Methanol, Diethylamine, 1,10-Phenanthroline	Chengdu KeLong Chemical Co., Ltd.	
Boric acid	Sichuan Xilun Chemical Co., Ltd.	
Acetonitrile	Thermo Fisher Scientific (China) Co., Ltd.	
Sulfuric acid (concentrated), Sodium hydroxide, Chloroform, Sodium bicarbonate	Chongqing Chuandong Chemical (Group) Co., Ltd.	Chongqing, China	

Experimental method

Culture and inoculation of microorganisms

Three strains of dominant phosphorus-solubilizing fungi were selected from the rhizosphere soil screenings of F. taipaiensis in the laboratory. A. tubingensis, A. niger, and A. fumigatus were inoculated with phosphorus- solubilizing fungal strains in the liquid medium shake flask culture. The inoculant was prepared by adjusting the concentration of the fungal suspension to 106 CFU/mL with sterile normal saline. Seven treatment groups and a control group (CK, with an equal volume of sterile normal saline as the control) were set up for the experiment. The treatment methods are shown in Table 2. Eight pots were used in each group with five plants/pot. The plants were inoculated once on March 10 and once on April 12. The F. taipaiensis planting base in Hongchiba Village, Wuxi County, Chongqing was routinely managed according to the F. taipaiensis cultivation method.

Table 2 Treatment groups and their inoculated strains.

Combination of strains and their respective dosages.

Treatment group	Inoculated strain	Inoculum size	
Z1	Aspergillus tubingensis	100 mL/Tub	
Z2	A. niger	100 mL/Tub	
Z3	A. fumigatus	100 mL/Tub	
Z12	A. tubingensis+A. niger	Each of the two strains is 50 mL/Tub	
Z13	A. tubingensis+A. fumigatus	Each of the two strains is 50 mL/Tub	
Z23	A. niger+A. fumigatus	Each of the two strains is 50 mL/Tub	
Z123	A. tubingensis+A. niger+A. fumigatus	Each of the three strains is 33 mL/Tub	

Sample collection

F. taipaiensis was harvested in June 2023 and the rhizosphere soil was extracted using the shaking root method. The herbs were dried in an oven at 35 °C, crushed through an 80-mesh sieve, sealed, and dried. Fresh rhizosphere soil was used to determine the abundance of microorganisms, and the remaining rhizosphere soil was naturally air-dried, crushed through an 80-mesh sieve, sealed, and dried.

Fresh and dry weights of the bulbs

The bulb samples were collected, cleaned, and air-dried to determine fresh weight, and then dried at 35 °C to determine dry weight.

Monomer alkaloids and total alkaloids of F. taipaiensis

About 2.0 g of F. taipaiensis fritillary sample powder was used to prepare a test solution according to a method outlined in the Chinese Pharmacopoeia (National PC, 2020). The absorbance value was determined using a visible spectrophotometer, and the content of total alkaloids in the sample was calculated according to a linear equation.

The reference products of peimine, peiminine, peimicine, sipeimine, and sipeimine-3β-D-glucoside were dissolved in methanol to prepare the gradient concentration reference solutions. The F. taipaiensis bulb samples were extracted by reflux to prepare the liquid for measurement. The reference solution and liquid to be measured were precisely absorbed and then detected using liquid-phase mass spectrometry (Zhou et al., 2014).

Quantitative determination of rhizosphere soil microorganisms

The dilution-coated-plate method was used to determine the abundance of microorganisms in the soil (Li et al., 2022). We weighed 1 g aliquot of fresh rhizosphere soil into a sterilized conical bottle, added 9 mL of sterile water, and placed the sample in a oscillating incubator at 180 r/min for 2 h. This was the 10−1 soil dilution soil suspension. Then, 1 mL of the 10−1 soil diluent was added to 9 mL of sterile water and the process was repeated, and subsequently on the 101–106 soil diluents. Bacteria and actinomycetes absorbed 100 µL of the 10−3–10−6 soil suspension in solid medium and fungi, and phosphorus-solubilizing fungi absorbed 100 uL of 10−2–10−5 soil suspensions in the solid medium, were coated evenly, and placed in an incubator for culture. Bacterial enumeration was conducted utilizing Luria-Bertani (LB) medium, which was incubated at 37 °C for a duration of 48 h. The quantification of actinomycetes was performed using a modified version of Gause’s No. 1 medium, incubated at 28 °C for 72 h. Fungal counts were executed on tetrazolium medium supplemented with chloramphenicol, also incubated at 28 °C for 72 h. Additionally, the assessment of phosphate-solubilizing fungi was carried out using an inorganic phosphate medium enriched with chloramphenicol, incubated under the same conditions of 28 °C for 72 h. During the counting process, only bacteria, actinomycetes, fungi, and phosphate-solubilizing fungi were enumerated, and morphological characteristics were used in conjunction with microscopic examination.

Determination of soil physicochemical properties

The soil physicochemical properties were determined using Bao’s (2000) soil agrochemical analysis. Soil pH was measured using a pH meter. Soil organic carbon and organic matter contents were determined by the potassium dichromate volumetric method and the dilution heat method, respectively. Soil available phosphorus was determined by sodium bicarbonate extraction and the colorimetric method. Soil total phosphorus was determined using the NaOH melt-molybdenum-antimony resistance colorimetric method. Soil available potassium was determined by the ammonium acetate extraction and atomic absorption method. Soil available nitrogen was determined by the alkaline hydrolysis diffusion method.

Data

WPS Office (Microsoft Inc., Redmond, WA, USA) was used for the statistics, SPSS 17.0 (SPSS Inc., Chicago, IL, USA) for analysis of variance, and Origin 21 (Origin Laboratories, Northampton, MA, USA) for mapping.

Results

Effect of phosphorus-solubilizing fungi on F. taipaiensis bulb biomass

The effects of different phosphorus-solubilizing fungi on bulb biomass of F. taipaiensis are shown in Table 3. The results indicated that the weight of the F. taipaiensis bulbs increased to different degrees compared with the CK group after inoculation with phosphorus-solubilizing fungi. Among them, the Z2 group inoculated with a single phosphorus-solubilizing fungus increased by 6.80% compared with the CK group. In the two groups inoculated with phosphorus-solubilizing fungi, the Z13 group had the largest increase of 51.14% compared with the CK group. In the group inoculated with three kinds of phosphorus-solubilizing fungi, the increase was 62.85% compared with that in the CK group. The moisture content of the bulbs was 68.07–70.43%. Groups Z13, Z23, and Z123 had the greatest effect on the biomass of F. taipaiensis. Therefore, the inoculation of phosphorus-solubilizing fungi significantly increased the bulb biomass of fritillary.

Table 3 Weight of bulbous fritillary bulbs before and after inoculation with phosphorus-solubilizing fungi.

No	Total fresh weight before planting/g	Total fresh weight after planting/g	Total dry weight/g	Gross weight gain/g	Weight gain rate/%	Drying rate/%	
Z1	9.4754	12.1728	3.8873	2.6974	28.47	31.93	
Z2	9.9037	12.9366	3.9734	3.0329	30.62	30.71	
Z3	9.6318	12.5259	3.8793	2.8941	30.05	30.97	
Z12	9.5762	11.9241	3.5355	2.3479	24.52	29.65	
Z13	9.8763	15.8126	4.7979	5.9363	60.11	30.34	
Z23	10.1577	19.522	6.0919	9.3643	92.19	31.21	
Z123	10.0342	18.7311	5.7588	8.6969	86.67	30.74	
CK	3.7351	4.6249	1.3676	0.8898	23.82	29.57	

Effect of phosphorus-solubilizing fungus inoculation on the total alkaloid content of fritillary bulbs

The effects of different phosphorus-solubilizing fungi on the total alkaloid contents of F. taipaiensis bulbs are shown in Fig. 1. After inoculation with the phosphorus-solubilizing fungi, the total alkaloid content of the F. taipaiensis bulbs in all treatment groups was significantly higher than that of the CK group (P < 0.05). Among them, the total alkaloid content of the Z3 group increased the most, followed by the Z12, Z13, and Z123 groups, which increased by 83.87%, 66.94%, 52.58%, and 43.55%, respectively, compared with the CK group, indicating that the quality of F. taipaiensis improved significantly after inoculation with phosphorus-solubilizing fungi.

Figure 1 Effect of different treatment groups on total alkaloids of F. taipaiensis bulbs.

Effect of phosphorus-solubilizing fungi on monomer alkaloids in F. taipaiensis bulbs

The effects of different phosphorus-solubilizing fungi on the monomer alkaloids from F. taipaiensis bulbs are shown in Table 4. The peimine content in all treatment groups was higher than that in the CK group, and the largest increase occurred in the Z123 group, with a growth rate of 101.51% compared with the CK group. The peiminine contents in the Z3 and Z23 groups were higher than that in the CK group, but the other groups were slightly lower than the CK group. The peimicine content in all groups except the Z12 group was higher than that in the CK group. The sipeimine content in all treatment groups was significantly higher than that in the CK group, and the largest increases in the Z3 and Z13 groups were 99.23% and 101.21%, respectively, compared with the CK group. The sipeimine-3β-D-glucoside content in the Z12 and Z23 groups increased by 45.94% and 46.25%, respectively, compared with the CK group. The total contents of the five monomer alkaloids are shown in Fig. 2. The total monomer alkaloid contents in all treatment groups except Z12 were greater than that in the CK group.

Table 4 Effects of the different treatment groups on fritillary bulb monomer alkaloids (μg/g).

No	Peimine	Peiminine	Peimisine	Sipeimine	Sipeimine-3β-D-glucoside	
Z1	27.597 ± 4.272bc	32.319 ± 4.938cd	371.347 ± 45.765ab	3.093 ± 0.495cd	1.216 ± 0.167abc	
Z2	29.604 ± 2.414b	30.632 ± 2.270d	353.026 ± 36.036ab	4.374 ± 0.439b	1.312 ± 0.101ab	
Z3	29.734 ± 5.310b	44.189 ± 6.900a	380.447 ± 44.902ab	5.193 ± 0.625a	1.063 ± 0.191cd	
Z12	23.240 ± 0.244c	36.385 ± 0.692bc	329.976 ± 6.086b	4.057 ± 0.166b	1.412 ± 0.113a	
Z13	27.567 ± 1.491bc	40.108 ± 2.508ab	394.749 ± 8.451a	5.245 ± 0.232a	1.075 ± 0.033cd	
Z23	27.573 ± 0.851bc	42.247 ± 1.049ab	363.253 ± 19.285ab	3.304 ± 0.128c	1.415 ± 0.018a	
Z123	38.802 ± 1.015a	37.183 ± 0.613bc	375.316 ± 5.857ab	4.296 ± 0.045b	1.181 ± 0.033bcd	
CK	19.255 ± 2.130d	41.927 ± 3.186ab	351.986 ± 40.417ab	2.607 ± 0.336d	0.968 ± 0.124d	

Figure 2 Effects of the different treatments on the total amount of monomer alkaloids in F. taipaiensis bulbs (μg/g).

Effects of phosphorus-solubilizing fungi on the rhizosphere soil microbial population of F. taipaiensis

The effects of inoculation with different phosphorus-solubilizing fungi on the abundance of microorganisms in the F. taipaiensis rhizosphere soil are shown in Fig. 3. Figure 3A shows that the rhizosphere soil bacteria in group Z3 had the highest number of colonies after inoculation with phosphorus-solubilizing fungi with 36.67 × 106 CFU/g, which was 382.46% higher than that in the CK group. As shown in Fig. 3B, the number of actinomycetes colonies in the fritillary rhizosphere soil was the highest in group Z23, which was 24.33 × 106 CFU/g, an increase of 234.11% compared with that in the CK group. As shown in Fig. 3C, the number of phosphorus-solubilizing fungal colonies in the F. taipaiensis rhizosphere soil group Z1 was the highest at 13.17 × 103 CFU/g, which was an increase of 27.43% compared with that in the CK group. According to Fig. 3D, the F. taipaiensis rhizosphere soil fungal colony number was lower in all groups (except Z12) compared to the CK group.

Figure 3 Effects of different phosphorus-solubilizing fungi on the number of microbes in F. taipaiensis rhizosphere soil.

(A) Bacteria, (B) actinomyces, (C) phospholytic fungi, (D) fungi.

Effects of different phosphorus-solubilizing fungi on F. taipaiensis rhizosphere soil physicochemical properties

The effects of inoculating phosphorus-solubilizing fungi on the physicochemical properties of the F. taipaiensis rhizosphere soil are shown in Fig. 4. Figure 4A shows that the pH value of the rhizosphere soil changed from 7.507 to 7.763 after inoculation with the phosphorus-solubilizing fungi. The soil pH value increased slightly after inoculation with phosphorus-solubilizing fungi compared with the CK group, but the overall change was small. As shown in Fig. 4B, rhizosphere soil-available phosphorus was significantly higher in all treatment groups than in the CK group. Available phosphorus in the CK group increased by 9.15% naturally, and by 127.77% and 86.48% in the Z123 and Z23 groups, respectively, compared with levels before cultivation (50.17 mg/kg). The increases were 108.68% and 70.85% compared with the CK group, respectively. Figure 4C shows that available potassium content in all groups increased to different degrees compared with the soil before cultivation, and the largest increases were in the Z13 group (504.08 mg/kg) and Z1 group (499.19 mg/kg), which increased by 129.49% and 127.26%, respectively, before cultivation. The increases in the CK group were 27.36% and 26.13% respectively. As shown in Fig. 4D, available nitrogen content in all treatment groups, except Z12, was higher than that in the CK group, and the maximum increase in available nitrogen content in the Z13 group (113.14 mg/kg) was 26.78% higher than in the CK group. As shown in Fig. 4E and 4F, total phosphorus (0.26%) and organic matter (74.14 g/kg) contents in the Z123 group were the highest, increasing by 38.50% and 37.92% compared with those in the CK group, respectively.

Figure 4 Effects of the treatment groups on rhizosphere soil nutrient content of F. taipaiensis.

(A) pH value, (B) available phosphorus, (C) available potassium, (D) available nitrogen, (E) total phosphorus, (F) organic matter.

Correlation analysis

The correlation analysis between alkaloid content and biomass, quantity of rhizosphere soil microbes, and F. taipaiensis soil nutrient content are shown in Fig. 5. The biomass of fritillary was positively correlated with available phosphorus and actinomyces (r = 0.943, 0.867, P < 0.01), and negatively correlated with phosphorus-solubilizing fungi and fungi (r = −0.829, −0.811, P < 0.05). Peimine was positively correlated with available phosphorus (r = 0.845, P < 0.01) and total phosphorus (r = 0.831, P < 0.05), and negatively correlated with fungi (r = −0.866, P < 0.01). A positive correlation was detected between peimicine and available nitrogen content (r = 0.934, P < 0.01), and a negative correlation was observed between peimicine and fungi (r = −0.714, P < 0.05). Available phosphorus was positively correlated with total phosphorus, organic matter, and actinomyces (r = 0.819, 0.764, 0.714, P < 0.05), and negatively correlated with phosphorus-solubilizing fungi and fungi (r = −0.777, −0.827, P < 0.05). Total P content was positively correlated with available P and organic matter (r = 0.819, 0.767, P < 0.05). Organic matter was positively correlated with available phosphorus content (r = 0.764, P < 0.05).

Figure 5 Correlation analysis between the indicators.

Note: P < 0.05: an asterisk (*) indicates correlation significant at the 0.05 level P < 0.01: two asterisks (**) indicate correlation significant at the 0.01 level.

Discussion

The effects of different treatments on biomass, alkaloids, and the rhizosphere soil environment of F. taipaiensis were studied by inoculating different phosphorus-solubilizing fungi in a pot experiment. The alkaloid content of F. taipaiensis reaches its highest level in 3–4 years and then decreases year by year, and the optimal harvest period is 4–5 years (Luo et al., 2021a). Therefore, the higher yield of F. taipaiensis produces higher economic value.

The results of this study indicate that, compared to the control group, inoculation with phosphate-solubilizing fungi increased the weight gain rate of the bulbs of F. taipaiensis by 2.94% (Z12) to 287.03% (Z23). Correlation analysis showed that the biomass of F. taipaiensis was significantly positively correlated with available phosphorus content; positively correlated with total phosphorus, organic matter, and available nitrogen; and negatively correlated with available potassium. The results of a study about the effects of combined application of nitrogen, phosphorus, and potassium fertilizers on the yield and total alkaloid content of Fritillaria unibracteata showed that the yield increase effect was, in order: phosphorus fertilizer > nitrogen fertilizer > potassium fertilizer (Deng et al., 2019), which was consistent with the results of this study. Therefore, this study confirms the hypothesis that inoculation with phosphate-solubilizing fungi is conducive to increasing the yield of F. taipaiensis, and the yield-promoting effect is significant in most treatments.

After inoculation with phosphorus-solubilizing fungi, the total alkaloid content of all treatment groups of F. taipaiensis bulbs increased by 6.35% to 84.08% compared to CK, showing a significant improvement. The alkaloid content reached the level stipulated in the 2020 Chinese Pharmacopoeia for F. taipaiensis (National PC, 2020) (>0.05%).

In terms of individual alkaloid content, inoculation with phosphate-solubilizing fungi generally increased the levels of peimine, sipeimine, and sipeimine-3-β-D-glucoside compared to the control group in the bulbs of F. taipaiensis. However, the content of Peimisine increased in most inoculated groups, with only a few groups showing a slight decrease. The changes in peiminine content were more complex, with most inoculated groups showing varying degrees of decrease, while a few groups showed an increase. This indicates that the effect of inoculating phosphate-solubilizing fungi on the synthesis pathway of peiminine is relatively complex.

Phosphorus content is closely related to alkaloid and secondary metabolism. Previous studies have shown that low phosphorus treatment can induce alkaloid synthesis in Pinellia ternata by regulating genes involved in phosphorus signal transduction and secondary metabolism, thereby increasing alkaloid content (Wang, 2022; Yao et al., 2024). However, since these studies were conducted under short-term low phosphorus conditions, changes in biomass were not observed. In this study, the alkaloid content in inoculated groups with high available phosphorus content (Z123, Z23, Z13) was higher than in those with lower phosphorus content (Z1, Z2, and Z12), which differed from previous results. This difference is mainly because inoculation with phosphate-solubilizing fungi affects not only phosphorus content but also increases available potassium and nitrogen levels. Therefore, whether low phosphorus content is more conducive to peiminine synthesis still requires further investigation. The decrease in peiminine content may be related to the significant increase in bulb biomass of Fritillaria taipaiensis promoted by fungal inoculation, meaning that peiminine synthesis did not increase simultaneously, leading to a decrease in its content. However, considering biomass analysis, the total yield of peiminine still increased. In group Z12, the contents of peiminine and peimisine significantly decreased, leading to a reduction in the total content of individual alkaloids; however, the total alkaloid content significantly increased, indicating a substantial rise in other alkaloids. Therefore, this study confirms the hypothesis that inoculation with Aspergillus phosphate-solubilizing fungi can increase the active ingredient content in F. taipaiensis.

Tang et al. (2021) observed that the synthesis of various monomer alkaloids in F. taipaiensis occurs concurrently with the production of peimine and peiminine, with a marginally greater concentration of peimine compared to peiminine. The findings of the present study align with these results. Peimine exhibits a significant positive correlation with both available phosphorus and total phosphorus, as well as a positive correlation with effective nitrogen and organic matter. This relationship may be attributed to the activity of soil enzymes, including urease and peroxidase (Chen et al., 2021). Conversely, it demonstrated a minimal correlation with available potassium. Sipeimine-3β-D-glucosides are positively correlated with the contents of available phosphorus, total phosphorus, and organic matter because organic matter is positively correlated with sipeimine-3β-D-glucosides (Ye et al., 2017), and strongly correlated with available phosphorus and total phosphorus (Tian et al., 2023). Peimisine, sipeimine, and total alkaloids exhibited a positive correlation with the levels of available potassium. In contrast, peimine, peiminine, and sipeimine-3-β-D-glucoside demonstrated minimal or negative correlation with available potassium. The precise effects and underlying mechanisms of these compounds warrant further investigation. Additionally, the concentration of peimisine showed a significant positive correlation with the availability of nitrogen.

In the treatment group inoculated with phosphate-solubilizing fungi, an increase in soil pH was observed in comparison to CK. However, existing literature indicates that the inoculation of phosphate-solubilizing fungi can lead to the secretion of organic acids during their growth, which may result in the dissolution of insoluble phosphorus in the soil and a subsequent decrease in soil pH (Zhao et al., 2024). This phenomenon may be attributed to the enhanced activity of phosphate-solubilizing fungi, which increases phosphorus availability in the soil, thereby facilitating greater phosphorus absorption by plants and mitigating the effects on soil acidity (Feng et al., 2017). A comprehensive analysis of the soil’s physicochemical properties suggests that the application of phosphate-solubilizing fungi can improve soil nutrient content. Furthermore, the efficacy of inoculating a combination of fungal strains appears to surpass that of individual strains, aligning with research findings regarding the growth-promoting effects of various strain combinations on Paris polyphylla var. yunnanensis (Xu et al., 2023).

Soil nutrient content affects the quantity of soil microorganisms and quality of fritillary, which is an important index of soil fertility. The number of bacteria increased significantly after inoculation with phosphorus-solubilizing fungi. A previous study reported that soil with higher insoluble phosphorus content was more conducive to increasing the number of rhizosphere soil bacteria. The correlation analysis showed that available phosphorus content was positively correlated with the number of bacteria (Wang, 2015). From the perspective of actinomycetes, the inoculation of a singular species of phosphorus-solubilizing fungi may result in a reduction in the population of actinomycetes. Conversely, the application of composite microbial fertilizers has the potential to enhance their numbers. Investigations into the diversity of actinomycetes within mangrove ecosystems, along with the antibacterial metabolites they produce, indicate that actinomycetes are capable of synthesizing a range of antibacterial compounds. These metabolites can influence the proliferation of other microorganisms, including phosphorus-solubilizing fungi, thereby indirectly affecting the population dynamics of actinomycetes (Xu et al., 2022). A solitary phosphorus-solubilizing fungus may engage in competition with actinomycetes for essential resources such as nutrients and spatial occupancy within the soil, potentially leading to a decline in actinomycete populations. In contrast, the presence of various strains within composite microbial fertilizers can facilitate the secretion of diverse organic acids, which are more effective in enhancing phosphorus-solubilizing capabilities, thereby indirectly fostering the growth of actinomycetes (Yin et al., 2023). From a correlational standpoint, an increase in the population of actinomycetes may lead to a corresponding rise in both total phosphorus and available phosphorus levels, a finding that aligns with prior research (Cao et al., 2023). The inoculated and purified phosphorus-solubilizing fungi significantly increased soil available phosphorus content. The number of phosphorus-solubilizing fungi decreased after the phosphorus-solubilizing fungi dissolved, thus increasing available phosphorus, total phosphorus, and organic matter contents in the soil. The fungi play phosphorus-dissolving and growth-promoting roles (Cao et al., 2023). Research indicates that as the number of consecutive cropping years increases, there is a linear decline in the populations of cultivable bacteria, actinomycetes, and the overall microbial biomass in the rhizosphere soil of Zhejiang Fritillaria. Conversely, the population of cultivable fungi exhibits a linear increase. This phenomenon results in a transition of the rhizosphere microbial community from a high-fertility “bacterial type” soil to a low-fertility “fungal type” soil. Furthermore, phosphate-solubilizing microorganisms not only enhance plant growth but also influence the abundance and community composition of rhizosphere soil microorganisms (Liao et al., 2011). In the present study, the inoculation of phosphate-solubilizing fungi led to a significant increase in bacterial populations within the rhizosphere soil of Fritillaria taipaiensis, while concurrently decreasing the fungal populations. In alignment with the nutrient classification criteria established by the second national soil survey, a positive correlation is observed between soil pH levels and soil nutrient content (Gu et al., 2020). In this study, the pH values of the treated groups were higher than those of the CK group, which may be related to the higher nutrient content. Therefore, applying antibiotics plays a role in regulating microecology and improving soil fertility.

Conclusion

In this study, the effects of inoculating phosphorus-solubilizing fungi on biomass and alkaloid content of F. taipaiensis were investigated. Inoculation with phosphorus-solubilizing fungi significantly increased the yield of F. taipaiensis. The biomass was positively correlated with the content of available phosphorus, total phosphorus, and organic matter, and the largest increase occurred in the Z123 group. After inoculation with phosphorus-solubilizing fungi, the total alkaloids of fritillary increased to different degrees compared with the CK group, and the largest increase was in the Z3 group. The contents of the four monomer alkaloids, except peiminine, were higher than those in the CK group, indicating that the quality of F. taipaiensis improved by adding phosphorus-solubilizing fungi. The F. taipaiensis rhizosphere soil microenvironment could be regulated after inoculation with phosphorus-solubilizing fungi. The Z23 and Z123 groups had the greatest influence on the number of bacteria and actinomycetes in the rhizosphere soil. The soil nutrient content in the Z13 group increased the most, and the contents of available phosphorus, available potassium, available nitrogen, total phosphorus, and organic matter increased by 47.71%, 27.36%, 26.78%, 25.13%, and 31.72%, respectively, compared with those in the CK group. The biomass of the bulb and alkaloid and soil nutrient contents were the best in the Z13 group. In summary, applying bacterial fertilizer to F. taipaiensis was highly beneficial and our results will provide a theoretical basis for the development and application of biological bacterial fertilizer.

Supplemental Information

Supplemental Information 1 Bulb fresh weight, dry weight, soil physical and chemical properties.

Supplemental Information 2 Total alkaloids of F. taipaiens.

Supplemental Information 3 Monomer alkaloid (chemistry).

We would like to express our gratitude to everyone who contributed to this research. Your efforts have ensured the smooth progress of the experiments and meticulous crafting of this article. We extend our deepest respect and appreciation for the hard work and significant contributions of each participant. Throughout the writing process, we used Wordvice AI for thorough grammar proofreading and to enhance the text, ensuring the language’s accuracy and clarity.

Additional Information and Declarations

Competing Interests

The authors declare that they have no competing interests.

Author Contributions

Xueyang Ren conceived and designed the experiments, performed the experiments, analyzed the data, prepared figures and/or tables, authored or reviewed drafts of the article, and approved the final draft.

Lin Yuan conceived and designed the experiments, performed the experiments, prepared figures and/or tables, authored or reviewed drafts of the article, and approved the final draft.

Huan Yao conceived and designed the experiments, performed the experiments, analyzed the data, prepared figures and/or tables, authored or reviewed drafts of the article, and approved the final draft.

Yuhan Wang performed the experiments, analyzed the data, prepared figures and/or tables, and approved the final draft.

Huan Wang performed the experiments, analyzed the data, authored or reviewed drafts of the article, and approved the final draft.

Dongqin Guo conceived and designed the experiments, authored or reviewed drafts of the article, and approved the final draft.

Nong Zhou conceived and designed the experiments, authored or reviewed drafts of the article, and approved the final draft.

Data Availability

The following information was supplied regarding data availability:

The bulb fresh weight dry weight, microbial population, soil physico-chemical properties; the total alkaloids of F. taipaiens, and monomer alkaloid content are available in the Supplemental Files.

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
