# Peer review of "Effects of phosphorus-solubilizing fungi on bulb quality and the Fritillaria taipaiensis rhizosphere soil environment"

_PeerJ, doi:10.7717/peerj.19283_

## Round 0.1 · original submission · Major Revisions

MS requires significant improvement in terms of language and methodological errors.

Reviewer 1 ·

Basic reporting

In a publication entitled "Effects of phosphorus-solubilizing fungi on bulb quality and the Fritillaria taipaiensis rhizosphere soil environment ", the authors focused on isolating fungal strains capable of solubilising phosphate and using them as a potential biofertiliser in the cultivation of the plant Fritillaria taipaiensis, which is an important part of natural medicine. The issue raised by the authors is of great interest because of the problem of plant utilisation of elements that are scarce in the soil, especially in the context of the diminishing supply of phosphate fertilisers.

The following is a review of the paper:

General comments:
The paper was prepared at a high level in many aspects. Especially in terms of graphics and tables. However, the paper needs many corrections on the linguistic side. There are many expressions in the work that are not used correctly, e.g:
Number of microorganisms -> should be: abundance of microorganisms
In the context of soil, the term cannot be used: bacterial suspension -> should be: soil suspension
The language used by the authors is also difficult to understand and needs detailed correction. Many sentences in the paper are unnecessarily long: According to the nutrient classification standard of the second National Soil Survey, the soil pH value is positively correlated with soil nutrients(Gu, et al., 2020). In this study, the pH values of the treated groups were higher than those of the CK group, which may be related to the higher nutrient content. Which could be shortened to the sentence -> In our study, the pH values of the treated groups were higher than those of the CK group, which may be related to the higher nutrient content, these results agree with the reports of Gu et al, 2020.
Other examples could be given. It is not the role of the reviewer to correct the language. It is up to the authors to check the article carefully and thoroughly in this aspect. And not just on the points mentioned by the reviewer. But the whole article.

Introduction:
In the introduction, the authors mention the release of phosphate by enzymes of the phosphatase group. However, this is not the only mechanism by which this element is released from soil. It is important to mention this in the introduction. Especially since the authors did not determine the activity of the phosphatases of the fungal strains studied in their paper.
The authors also describe in very general terms the strains involved in the release of phosphate in soil. They do not give examples of species that have already been studied and described in this aspect. In particular, examples from the genus Aspergillus are not given.

Furthermore, the full genus name, Aspergillus, is only given for the first time in Table 1. The full genus name should be used the first time the name appears in the text. That is, for authors in the abstract. This should be corrected.

Results.
This chapter mainly needs to be corrected in terms of the descriptions provided. The authors use long, convoluted descriptions, comparing each parameter obtained with a control sample (CK). This is very difficult for the reader. The authors need to extract the most important information from their results and point out the most important test systems, giving the most significant results. This applies to each subsection in this chapter.
Lines 164-173 - the species name is missing italics.
Figure 3 - change number to abudance
Table 3. - There was an error in the formatting of study group Z123 in line 7.

Discussion:
The authors describe the concentration of alkaloids in the studied plant. It would be interesting to know whether there are any reports in the literature on the dependence of the synthesis of these compounds on the availability of phosphorus? Not necessarily using the example of the plant studied, but others.
The authors point out that after inoculation with fungi in the Z13, Z23 and Z123 configurations, the overall abundance of phosphate-solubilising fungi decreases in favour of the abundance of actinomycetes. This is a very interesting change. Are the authors able to place their results in the context of the literature?

Experimental design

Methodology:
Study material - Key question. Why did the authors use sterilised soil for the study? Especially in the context of microbiological studies? It is obvious that in order to achieve complete sterility of the soil, it is necessary to cauterise it 3 times at 24 hour intervals. However, I do not understand why such soil was used in experiments ?
Experimental Drugs - The name of this chapter suggests the use of drugs. However, it describes the substrates, reagents and standards used in the experiments. The name of the chapter should be changed to, for example, Chemicals and Ingredients.
In addition, if reagents are used, their origin should also be stated: manufacturing company, country.
Line 106-108 - The sentence in this line suggests that the isolate under study was inoculated with other fungal strains capable of solubilising phosphates?
Line 109 - I think it was a fungal suspension? Not bacterial?
The most important flaw in this chapter is the complete lack of information on the origin of the isolates tested. Where they were isolated from. How they were identified. On what basis they were selected for study. If the research on these strains has been described in another publication, this should be cited. If this is the first report of these strains. This information should be included here. Absolutely.
Fresh and dry bulb weights - how long were the samples dried to obtain dry weights? Why did the authors air dry the samples before determining the wet weight?
Line 127 - just write 2 g or 2.0 g, no need to write 2.0000 g
Quantitative determination of rhizosphere soil microorganisms - this section raises the most questions. Were the microbial counts done on selective media? Were antibiotics added to ensure growth of only certain groups of microorganisms? This is important because a number of fungal strains will grow freely on bacterial media. More detailed information is needed on the types of media used.

Validity of the findings

no comment

Reviewer 2 ·

Basic reporting

The fungi strains used in the paper should be described in detail in the Introduction

The hypothesis that the authors tested during the experiments is not formulated. It would be good to do this in the Introduction, and in the Conclusions section, to describe whether the hypothesis was confirmed or refuted.

Experimental design

The source from which the fungal strains were obtained should be described in detail in the Methods section.

Validity of the findings

The Introduction section should describe in detail the authors' motivation to study these biochemical parameters of the bulbs.

The figure captions should be more detailed and describe what the lowercase letters a-h in Figures 1, 3, and 4 mean, as well as their combinations—ABC, cd, etc.

It is not clear from the diagram in Figure 5 what "fungus" means

Table 2 gives the data on changes with 4 and 2 decimal places. Were such precise weighings carried out?

A similar remark to the data in Table 3. Is it necessary to use 3 decimal places? What do bc, b, cd, and other lowercase letters mean?

Additional comments

A. tubingensis and other species should be deciphered in the Abstract and first used in the text. In its present form, the designation A. appears only in Table 1.

---

## Round 0.2 · Minor Revisions

Authors have addressed all the reviewer comments.

Curtis Daehler, the Section Editor, has commented and said:

"This manuscript is well-written but I think the authors need to check the following issues:

1. L 45 and concluding statement at the end of the manuscript "Z13 group was the best treatment group when considering bulb biomass and alkaloid and soil nutrient contents." - please check this statement -- from the graphs and tables it looks like Z123 had higher bulb biomass and higher soil phosphorus than Z13.

2. L 50 - please write out Fritillaria at the start of the Introduction rather than starting the sentence with the abbreviation "F."

3. L 160 "vaccinated" should say "inoculated".

4. L 358 "Peimine compared to Peimine" (these are identical, I think one should be "Peiminine").

5. L 417 " Fritillaria thunbergii" should be in italics.

6. Table 3 - The Z123 treatment is missing and the Z12 treatment is duplicated. I think the second Z12 should be Z123 .

7. Table 4 - A strange small symbol like a dot appears 3 time causing mis-alignment of numbers in Table 4.

8. L 32 Consider deleting "The results showed that", instead beginning with "Inoculation..."

9. L 18-22 - It is good to use the species authority "P.Y. Li" at first on L 18, but repeating the species authority is not typically needed in the subsequent instances of the species name."

Please address these so we can move forward.

Reviewer 1 ·

Basic reporting

The authors of the submitted work have taken into account all the comments of the reviewer and have significantly improved the quality of the work. The work has become more readable and the authors have corrected many shortcomings. They have also clarified ambiguities in the methodology and broken it down well. However, I would suggest an editorial correction before publication, as italics are often missing in sentences. For example, in lines 24, 224 and 1313. Also, in line 584 there is a capital letter in the middle of the sentence. This should be corrected.

Experimental design

no comment

Validity of the findings

no comment

Additional comments

no comment

Reviewer 2 ·

Basic reporting

The authors have significantly improved the manuscript

Experimental design

no comments

Validity of the findings

no comments

Additional comments

no comments

---

## Round 0.3 · accepted · Accept

Authors have addressed all the comments. The manuscript can be accepted for publication.